# Three-Stage Tone Mapping Algorithm

Lanfei Zhao [1] , Ruiyang Sun [2] and Jun Wang [3],*

1    The Higher Educational Key Laboratory for Measuring & Control Technology and Instrumentations of Heilongjiang Province, Harbin University of Science and Technology, Harbin 150080, China
2    Chinese Martial Arts Department, Harbin Sport University, Harbin 150008, China
3    School of Information Engineering, Quzhou College of Technology, Quzhou 324000, China
*    Correspondence: 145688@qzct.edu.cn; Tel.: +86-181-0570-1939

**Abstract:** In this paper, a tone mapping algorithm is presented to map real-world luminance into displayed luminance. Our purpose is to reveal the local contrast of real-world scenes on a conventional monitor. Around this point, we propose a three-stage algorithm to visualize high dynamic range images. All pixels of high dynamic range images are classified into three groups. For the first stage, we introduce piecewise linear mapping as the global tone mapping operator to map the luminance of the first group, which provides overall impressions of luminance. For the second stage, the luminance of the second group is determined by the weighted average of its neighborhood pixels, which are derived from the first group's pixels. For the third stage, the luminance of the third group is determined by the weighted average of its neighborhood pixels, which are derived from the second group's pixels. Experimental results on several real-world images and the TMQI database show that our algorithm can improve the visibility of real-world scenes with about 12% and 9% higher scores of mean opinion score and tone-mapped image quality index than the closest competitive tone mapping methods. Compared to the existing tone mapping methods, our algorithm produces visually compelling results without halo artifacts and loss of detail.

**Keywords:** high dynamic range; tone mapping; piecewise linear mapping; local contrast

## 1. Introduction

The real-world scene covers a wide range of luminance levels, whose order of magnitude approximates about $10^{14}$:1 [1]. Through the complicated process of self-adaption, the human visual system adapts over nine orders gradually. The dynamic range of modern image sensors is far less than real-world scenes and human visual systems. Nevertheless, by multi-exposure imaging [2], the limit of the image scene can be extended. Hence, we can obtain high dynamic range (HDR) images with conventional digital cameras.

The contrast ratio of some advanced conventional displays can reach up to $10^4$:1 [3], although some high dynamic range display systems claim that they can achieve five orders of magnitude [4]. However, the dynamic range of display systems is much less than real-world scenes. Furthermore, the price of high contrast ratio displays is not affordable for household consumers. This flaw results in shadow and highlight that can not convey the appearance of the original scene. Tone mapping has been developed to convert an HDR image to a low dynamic range (LDR) image, whose dynamic range is compatible with conventional displays. A valid tone mapping algorithm can reveal both darkness and brightness pixels clearly on a conventional display. It is also able to preserve details in the original image and avoid common artifacts, such as halos, gradient reversals, or loss of local contrast [5].

Unfortunately, existing tone mapping algorithms suffer from some reduction in image quality, e.g., detail loss and unsuitable overall impression of brightness. To solve the rendering problem of HDR images, we present a three-stage tone mapping algorithm. The main idea is to map real-world luminance to displayable luminance while preserving

local contrast in the real-world scene. For mapping real-world luminance, previous tone mapping methods compress mainly high luminances [6]. Such tone-curves suffer from over-compression of the global contrast. To deal with this problem, we propose a piecewise linear mapping as the first stage to determine the brightness and enhance the global contrast of tone-mapped images. Each segment point of the piecewise linear mapping is selected by a tiny luminance threshold, and the remapped luminance is estimated by the cumulative probability of the luminance histogram. For visual detail protection, Ashikhmin [7] focused on preserving local contrast during compressing dynamic range. However, such a method could lead to insufficient performance of visual detail preservation. We introduce another two stages to ensure local contrast; both stages employ the weighted average of neighborhood pixels to estimate local contrast. The luminances derived from each stage are related to the previous stage by reviewing the local contrast expression. Experiments were conducted on radiance maps from various real-world scenes and show that the proposed algorithm can produce pleasing images.

## 2. Related Work

In an early attempt, global tone mappings designed a single curve to compress dynamic range and was designed to save computational efficiency [8]. A practical general framework to match real-world brightness and display brightness was built by Tumblin and Rushmeier [9]. They reviewed the sigmoid responses to light in the film's encoding process. Using this kind of power-law relation, they created an observer model that converts world luminance to perceived brightness; meanwhile, the coefficient and exponent are determined by Stevens's experiment. Ward [10] applied a designated multiplier to relate the minimum discernible difference on the display and in a real-world scene. Note that this method deals with the perceived contrast, consequently, it provides the same contrast visibility both in bright scenes and in dark scenes. Larson et al. [11] suggested that the eye is sensitive to region brightness, i.e., adaptation levels, rather than absolute luminance. To estimate adaptation levels, they averaged real-world luminance over a 1○ visual angle. Histogram equalization with a linear ceiling was applied to compress the dynamic range while the limit region contrast could not exceed the original image. Khan et al. [12] presented a histogram-based tone mapping algorithm to visualize HDR images on an LDR display. The algorithm restricts the pixel counts in the histogram to solve over-compression and over-enhancement problems introduced by classic histogram equalization.

Recent tone mapping research focuses on local tone mapping operators. Meylan et al. [13] rendered real-world images with a Retinex-based adaptive filter. This filter processes the luminance and chrominance in two parallel procedures. The chrominance processing employs principal component analysis to provide unaffected color rendition immunity while luminance is compressing. Gu et al. [14] developed a local edge-preserving (LEP) filter and applied the LEP filter to decompose real-world scene images into several detail layers and one base layer based on the Retinex model. Kuang et al. [15] proposed a valid image appearance modeling to reproduce the same visual perception across media based on the iCAM framework. This model decomposes an image into a base layer and a detail layer by the bilateral filter. The Cone response function and the rod's response function are applied to the base layer to compress the dynamic range. A power-function adjustment is applied to detail layer to predict the Stevens effect. Fattal et al. [5] took the luminance gradient as the local metric to preserve details and compress the dynamic range. The work attempts to attenuate the large magnitudes of gradients at each pixel while small magnitudes of gradients remain unchanged. Their work also introduces a Gaussian pyramid to avoid halo artifacts.

In recent years, deep-learning-based tone mapping algorithms have been presented to generate LDR images. Rana et al. [16] proposed a fast, parameter-free, and scene-adaptable deep tone mapping operator (DeepTMO) based on a conditional generative adversarial network. DeepTMO explores four possible combinations of Generator-Discriminator architectural designs to specifically address some prominent issues in HDR related deep-learning

frameworks like blurring, tiling patterns, and saturation artifacts. Panetta et al. [17] designed a deep-learning-based tone mapping operator (TMO-Net), which offers an efficient and parameter-free method capable of generalizing effectively across a wider spectrum of HDR content. Patel et al. [18] proposed a novel generative adversarial network to learn a combination of several classic tone mapping operators. The proposed method uses a deep network to keep the best TMQI [19] score tone-mapped image generated by the classic tone mapping operators.

## 3. Algorithm

### 3.1. Global Luminance Mapping

We apply the scene's key value proposed by Reinhard [20] to make an analogy to camera exposure. The approximation to the key of the scene is given by

$$\overline{L} = \frac{1}{M} \exp\left( \sum_{x,y} \Big( \log(\delta + \widetilde{L}_{x,y}) \Big) \right) \tag{1}$$

where $M$ is the total number of pixels, $\delta$ is a small value, $\widetilde{L}$ is world luminance, $(x, y)$ is the pixel location and $\overline{L}$ is the key of the scene. The scaled luminance is computed by

$$L_{x,y} = \frac{a}{\overline{L}} \times \widetilde{L}_{x,y} \tag{2}$$

where $a$ is the key value to map the log-average world luminance to an appropriate value and $a$ is estimated adaptively by Reinhard's work [21].

We found that the tone distribution curve of a great number of radiance maps presents a similar shade, partially shown in Figure 1a, which is obtained from a well-known radiance map called "memorial church".

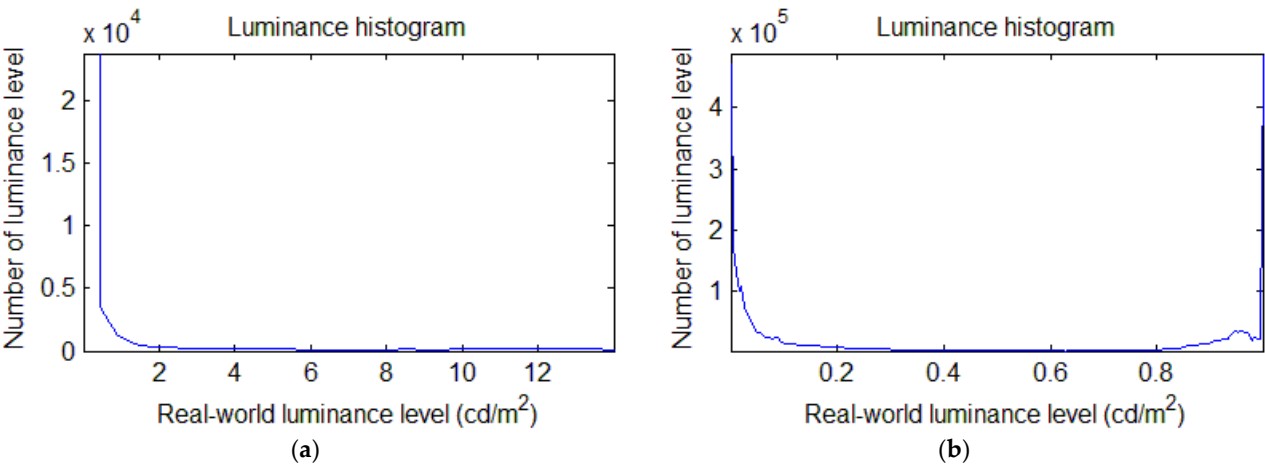

**Figure 1.** Luminance histogram of memorial church image. (**a**) Memorial Church; (**b**) Moto.

It can be seen that most luminance levels distribute at the beginning of the luminance level, and the curve plots a very long and low trailing. This indicates that it is necessary to make the compression for high luminance levels. Meanwhile, a majority of luminance levels, i.e., the low luminance levels, should be stretched to increase the global contrast.

Not all of the luminance histograms look like Figure 1a. The luminance levels of some scenes occur in both low-level bins and high-level bins, as shown in Figure 1b. This histogram comes from another radiance map called "moto". Here, we can see that it has equal importance both at low-level and high-level bins. This type of curve suggests that low-level and high-level bins should be stretched together.

Essentially, to pick a luminance mapping operator is to choose a proper shade of the mapping curve. If employing linear mapping as the luminance mapping, the visual of the remapping image is supposed to be consistent with the reconstructed image by the optic

nerve practically. However, the linear mapping is incapable of compressing dynamic range within the range of HDR images. Due to this, a piecewise linear mapping is advisable to replace linear mapping.

The next issue is to determine segment points for piecewise linear mapping. Since the bins possess very few numbers of luminance levels that should be compressed, we use the empirical value $T_a = 10^{-4}$ as the threshold to determine the location of segment points. When the probability of the previous bin is larger than the empirical value, while the probability of the current bin is not greater than the empirical value, the current bin is a segment point. When the probability of the previous bin is not greater than the empirical value, while the probability of the current bin is larger than the empirical value, the current bin is also a segment point. Moreover, the remapping value of a segment point is determined by the cumulative probability between the previous segment point and the current segment point. Let $B(\cdot)$ represent the luminance bin and $(\cdot)$ represent the number of a bin. The remapping value of the *l*-th bin between two segment points $B(i)$ and $B(j)$ is given by

$$L(B(l)) = \frac{L(B(j)) - L(B(i))}{B(j) - B(i)} \times (B(l) - L(B(i))) + L(B(i)) \qquad (3)$$

where $L(B(l))$, $L(B(i))$, and $L(B(j))$ are the remapping value of the *l*-th bin, *i*-th bin, and *j*-th bin, respectively. Note that the remapping values of the first and last bin are 0 and 1, respectively. Thus, the remapping value of every segment point is known by formula (3). Using (3) after segmenting the histogram by an empirical threshold, we obtain the luminance mapping curves shown in Figure 2. Due to the slow varying specifically at the end of the curve shown in Figure 1a, the piecewise linear function mainly compresses the range of high luminance for memorial church image. Meanwhile, the low luminance level is stretched to enhance the contrast in dark areas. Both low luminance levels and high luminance levels are stretched and the dynamic range of the middle luminance level is compressed since most bins are densely distributed both at the beginning and end of the luminance histogram which is shown in Figure 1b.

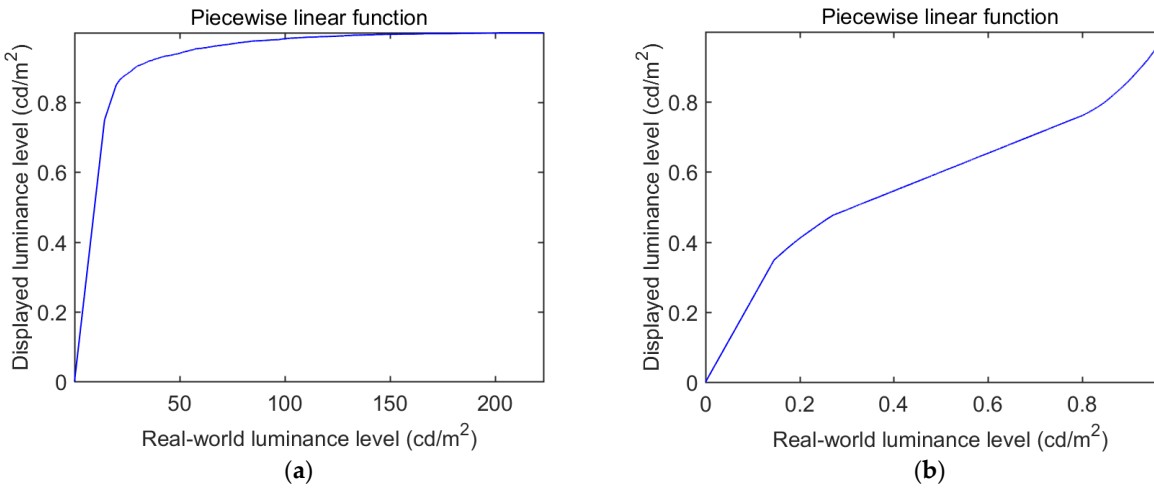

(**a**)  (**b**)

**Figure 2.** Piecewise linear function. (**a**) Memorial Church; (**b**) Moto.

### 3.2. Local Contrast Preservation

To preserve local contrast, all pixels of the input image are divided into three groups, i.e., $U_1$, $U_2$, and $U_3$. Figure 3 shows a classification procedure for an image with a size of $9 \times 9$. Both row and column numbers of the first group's pixels are odd, as shown in Figure 3a. There is only one spatial coordinate of the second group's pixels that is odd, as shown in Figure 3b. Both row and column numbers of the third group are even, as shown in Figure 3d. The second group pixel is divided into two subgroups, as shown in Figure 3c.

The row number of the first subgroup is even. On the contrary, the row number of the second subgroup is odd.

**Figure 3.** Pixel classification. (**a**) Pixels of group one; (**b**) Pixels of group two; (**c**) Subgroups for group two; (**d**) Pixels of group three.

We use the global luminance mapping described in Section 3.1 to compress the dynamic range for the first group's pixel. Then, the local contrast of the two subgroups is computed by

$$
\begin{cases}
c_{x,y}^{(1)} = \dfrac{3I_{x,y}}{I_{x-1,y}+I_{x,y}+I_{x+1,y}}, & (x,y) \in U_2^{(1)} \\
c_{x,y}^{(2)} = \dfrac{3I_{x,y}}{I_{x,y-1}+I_{x,y}+I_{x,y+1}}, & (x,y) \in U_2^{(2)}
\end{cases}
\tag{4}
$$

where $(\cdot)$ represents the number of subgroups, $c$ represents local contrast, $I$ represents displayed luminance, and the superscripts of $c$ and $U$ represent the number of the subgroup. For local contrast preservation, the local contrast value of every pixel should be equal to the tone-mapped image, which can be formulated as:

$$
\begin{cases}
c_{x,y}^{(1)} = \dfrac{3L_{x,y}}{L_{x-1,y}+L_{x,y}+L_{x+1,y}}, & (x,y) \in U_2^{(1)} \\
c_{x,y}^{(2)} = \dfrac{3L_{x,y}}{L_{x,y-1}+L_{x,y}+L_{x,y+1}}, & (x,y) \in U_2^{(2)}
\end{cases}
\tag{5}
$$

Since the luminance of the first group's pixels has been computed by global luminance mapping, i.e., $L_{x-1,y}$, $L_{x+1,y}$, $L_{x,y-1}$, and $L_{x,y+1}$ are known, the luminance of the second group's pixels derived by joint (4) and (5) is computed by:

$$
I_{x,y} =
\begin{cases}
\dfrac{c_{x,y}^{(1)} \times (L_{x-1,y}+L_{x+1,y})}{3-c_{x,y}^{(1)}}, & (x,y) \in U_2^{(1)} \\
\dfrac{c_{x,y}^{(2)} \times (L_{x,y-1}+L_{x,y+1})}{3-c_{x,y}^{(2)}}, & (x,y) \in U_2^{(2)}
\end{cases}
\tag{6}
$$

The local contrast of the third group's pixels is computed by:

$$
c_{x,y} = \frac{5I_{x,y}}{I_{x-1,y} + I_{x+1,y} + I_{x,y} + I_{x,y-1} + I_{x,y+1}}
\tag{7}
$$

Similar to the second group's pixels, the luminance of the third group's pixels is computed by:

$$
I_{x,y} = \frac{c_{x,y}^{(3)} \times \left(L_{x-1,y} + L_{x+1,y} + L_{x,y-1} + L_{x,y+1}\right)}{5 - c_{x,y}^{(3)}}
\tag{8}
$$

Figure 4 shows the flowchart of the three-stage approach.

All the pixels of the input real-world image are first classified into three groups. In Stage 1, the luminances of the pixels in group one are remapped by piecewise linear mapping. Both Stage 2 and Stage 3 compute the local contrast of every pixel of group two and group three. After piecewise linear mapping, i.e., Stage 1, Stage 2 computes the

luminance of all the pixels in group two, according to Equation (6). After Stage 2, Stage 3 computes the luminance of all the pixels in group three, according to Equation (7). All pixels of the three groups are collected to generate the tone mapped image.

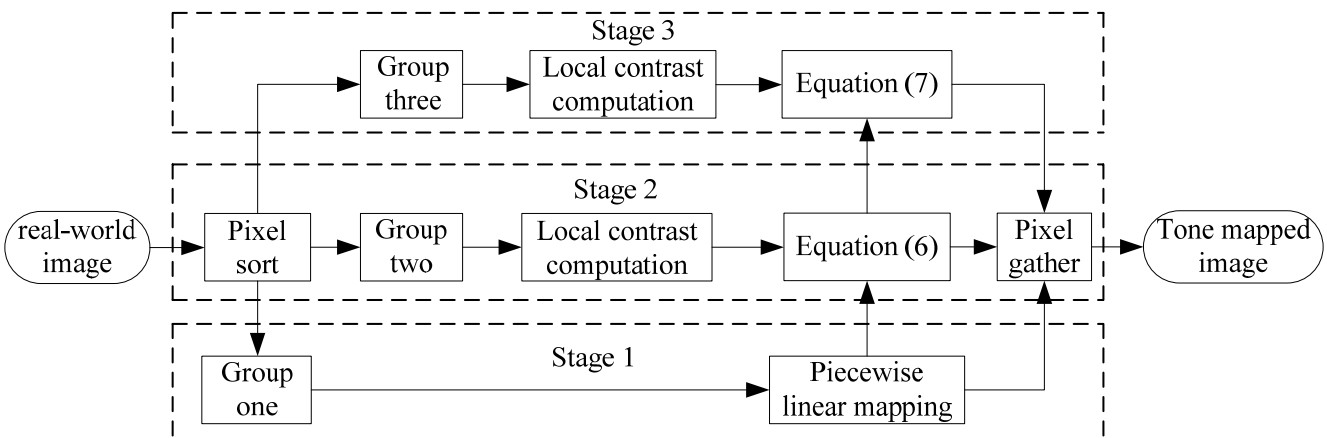

**Figure 4.** The flowchart of the three-stage algorithm.

### 3.3. Color Image

We separate lightness and color features by converting RGB images into HSV space. Figure 5 shows the data flow diagram for dealing with a color image.

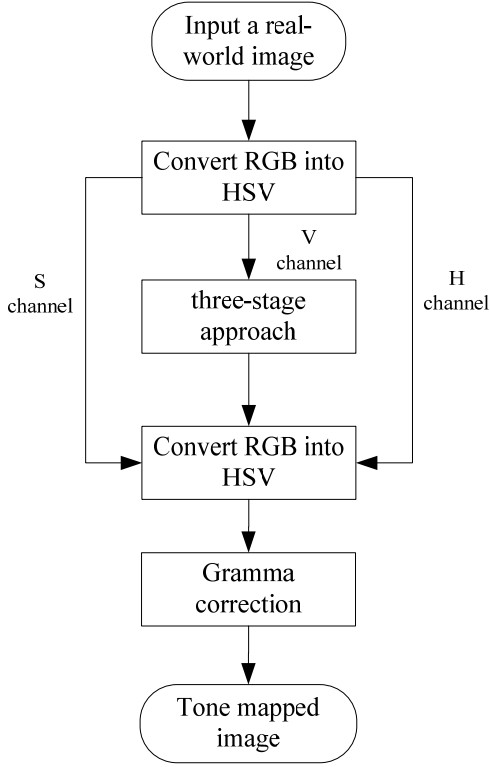

**Figure 5.** Data flow diagram for dealing with a color image.

Three channels are separated into H, S, and V channels, where the V channel is only used to compress dynamic range, and the H and S channels are constant. The last part is Gamma correction, which is used for overcoming the non-linear response of the display device.

## 4. Results

Our method is simulated by Matlab 2016a on a desktop computer with an Intel i5-7400 CPU and Kingston 2400 MHz 8G-DDR4 internal storage. Tone-mapped images are shown on a Dell S2721DGF digital display whose gamma parameter approximates 2.2. The test radiance maps contain three famous scenes and the TMQI database [19] with a very high dynamic range.

### 4.1. Subjective Performance Evaluation

In this part, we use three famous radiance maps to make a comparison among several tone mapping algorithms. The compared tone mapping algorithm include DeepTMO [16], Thai [22], TMO-Net [17], L1-L0 [23], and Khan [12]. The three test radiance maps are shown directly in Figure 6. Figures 7–9 demonstrate a comparison of tone-mapped images.

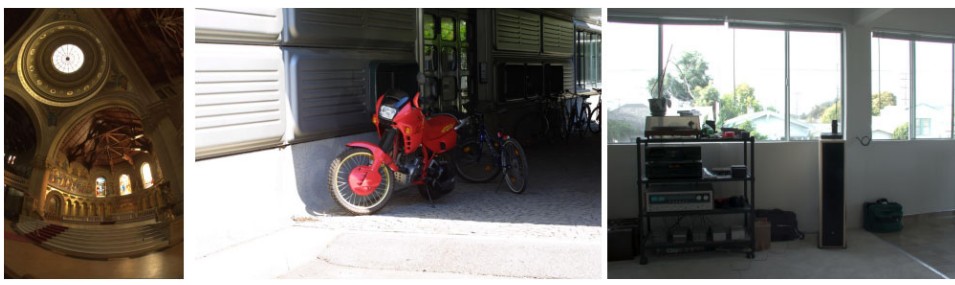

**Figure 6.** The picture shows the test radiance maps rendering on standard LCD. The left image, called "memorial church", is an HDR image from TMQI database [19]. The middle image, called "moto", and the right image, called "apartment", are reprinted with permission from Ref. [24]. 2022 Elsevier B.V. (Reprinted from Signal Processing: Image Communication, Vol (29), Authors (Narwaria, M., Da Silva, M.P., Le Callet, P. and Pépion, R.), Title of article (Tone mapping based HDR compression: Does it affect visual experience?), Pages (257–273), Copyright (2013), with permission from Elsevier).

The images derived by our algorithm show visually pleasing results due to the clear details and distinguishable information both in dark areas and bright areas. The compared tone mapping algorithms suffer from some reduction in image quality. TMO-Net provides a good, clear result in the darker areas, while the contrast in the brighter area is decreased which leads to a problem of overexposure. DeepTMO improves the global contrast of the original real-world image. The content of the dark area is revealed and the brightness becomes more apparent. However, the reproduced image was blurred while the details are reduced both in low-light and bright areas. Both Thai and Khan have the problem of contrast loss in the low-light area, which leads to over-compression of some intensity levels. There, some image quality is lost, particularly in darker areas. The visibility of the dark area is improved by L1-L0 and the tone-mapped images make a good balance between detail enhancement and visual naturalness for both indoor and outdoor scenes. However, the LDR image obtained by L1-L0 has problems with halo artifacts and the contrast is over-enhanced. Meanwhile, the tone-mapped images show a little brightness distortion.

Another experiment was conducted to test the validity of the tone mapping algorithm. We selected the TMQI database [19] that contains 15 radiance maps to compute the mean opinion score (MOS). TMQI database is shown in Figure 10. It can be seen that the TMQI database includes six indoor scenes and nine outdoor scenes. Due to the unequal resolution, we resize the images without aspect ratio preservation to favor the demonstration. The image displayed in Figure 10 leads to unsatisfactory visibility which is caused by extremely high contrast. The algorithms compared and the algorithm proposed were adopted to convert every radiance map of the TMQI database to a displayed image. Thus, we obtained 90 LDR images. Figure 11 shows four groups of LDR images. It can be seen from Figure 11 that the visibility of our method is improved with better performance than the other competitive tone mapping methods. The subjective visual quality of each tone-mapping image share quality consistent with the reproduced image shown in Figures 7–9.

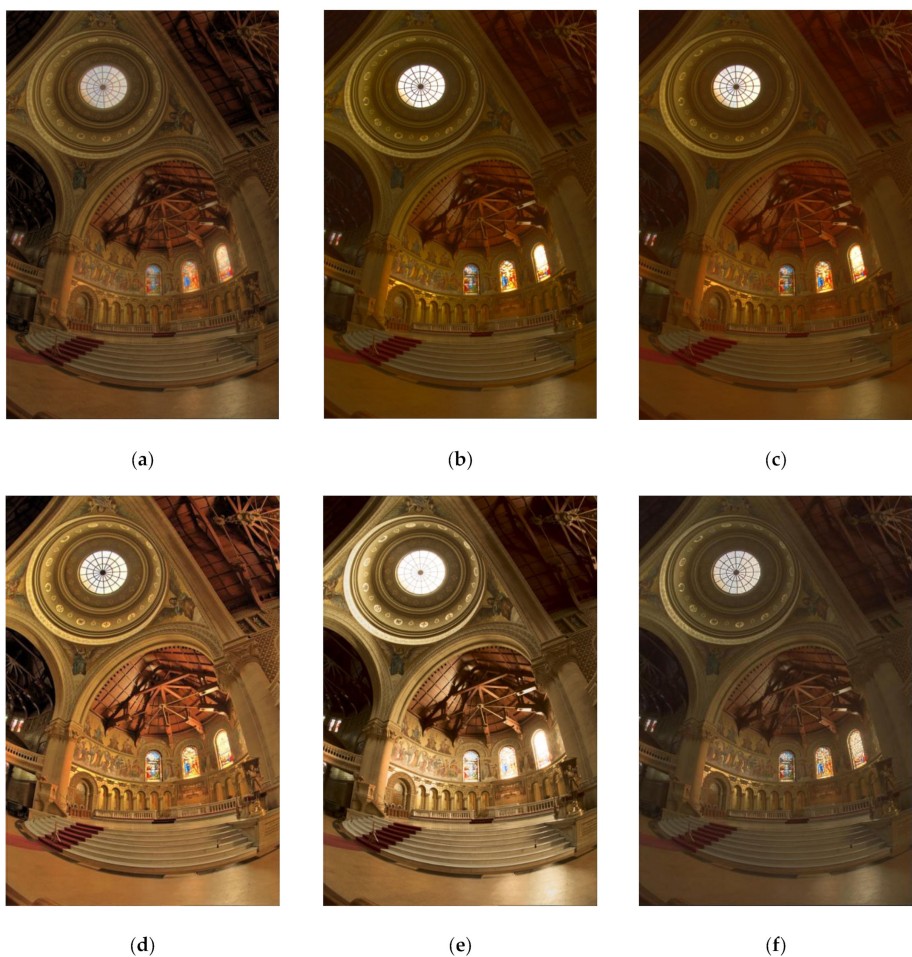

**Figure 7.** A comparison of tone mapping algorithms on the memorial church image. (**a**) Result by TMO-Net. (**b**) Result by DeepTMO. (**c**) Result by Thai. (**d**) Result by L1-L0. (**e**) Result by Khan. (**f**) Result by our algorithm.

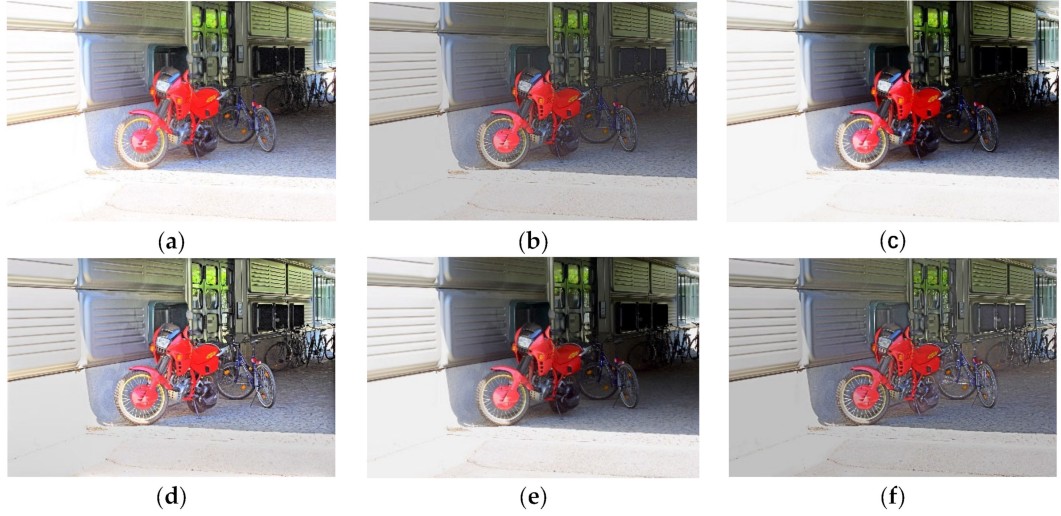

**Figure 8.** A comparison of tone mapping algorithms on the moto image. (**a**) Result by TMO-Net. (**b**) Result by DeepTMO. (**c**) Result by Thai. (**d**) Result by L1-L0. (**e**) Result by Khan. (**f**) Result by our algorithm.

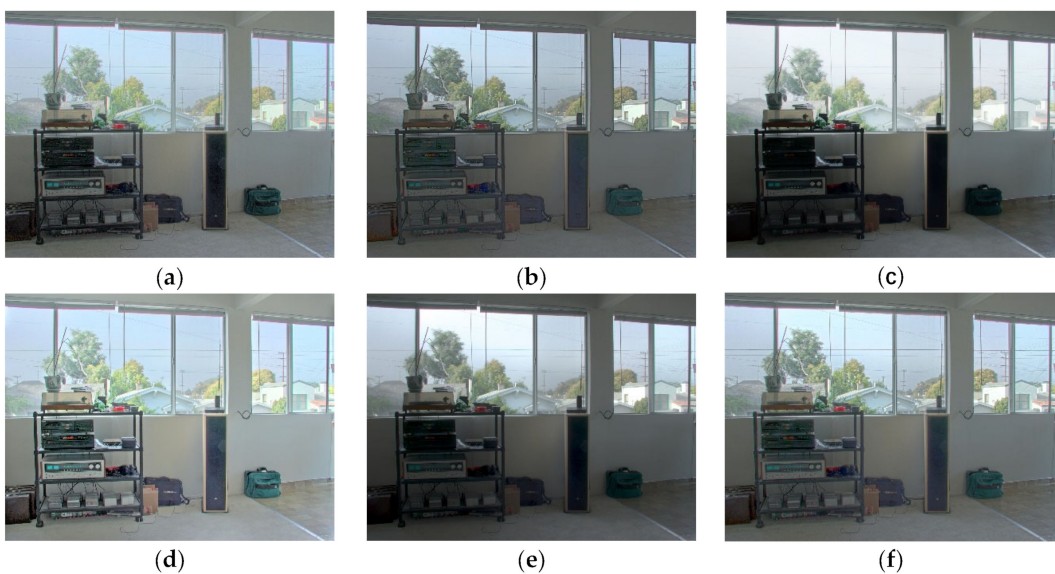

**Figure 9.** A comparison of tone mapping algorithms on the apartment image. (**a**) Result by TMO-Net. (**b**) Result by DeepTMO. (**c**) Result by Thai. (**d**) Result by L1-L0. (**e**) Result by Khan. (**f**) Result by our algorithm.

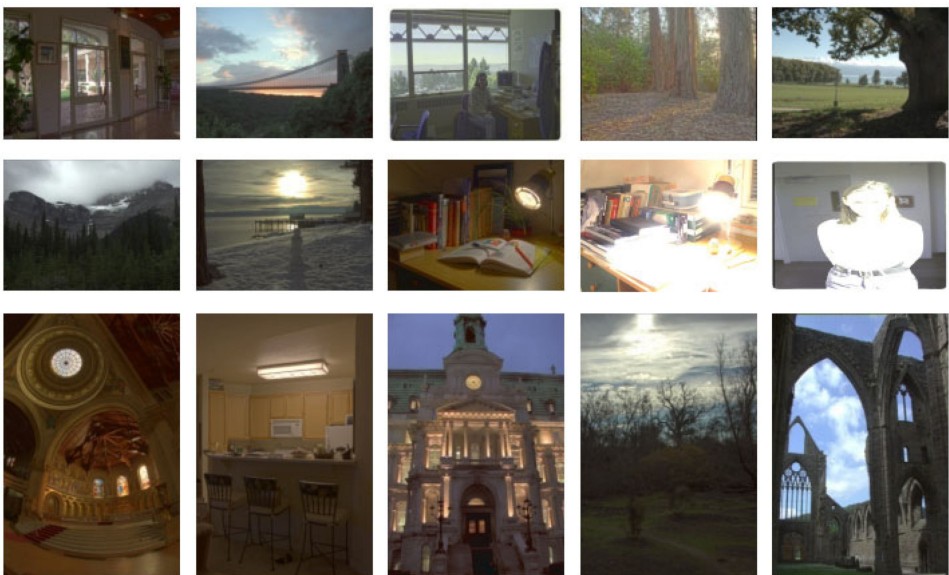

**Figure 10.** The 15 radiance maps from the TMQI database.

Then, the tone-mapped images of the TMQI database were shown on an S2721DGF Display. To determine MOS, 31 volunteers, including 16 females and 15 males aged between 23 and 34, were invited to give the score for each image shown on the display produced by the above tone mapping algorithms. The score range is from 1 to 10, where 1 means the worst visual quality and 10 means the best visual quality. The mean and std of MOS values are shown in Figure 12.

It can be seen that our method has a higher score (8.9) than other compared methods. Meanwhile, the lowest std value (0.175) means the performance of our algorithm is widely recognized among these volunteers. The mean score and standard deviation for other tone mapping methods are DeepTMO (7.9, 0.3), Thai (5.8, 0.6), Khan (5.1, 1.1), TMO-Net (7.2, 0.25), and L1-L0 (7.7, 0.4). According to the mean scores, our method achieves about 12% higher than the closest tone mapping method (DeepTMO).

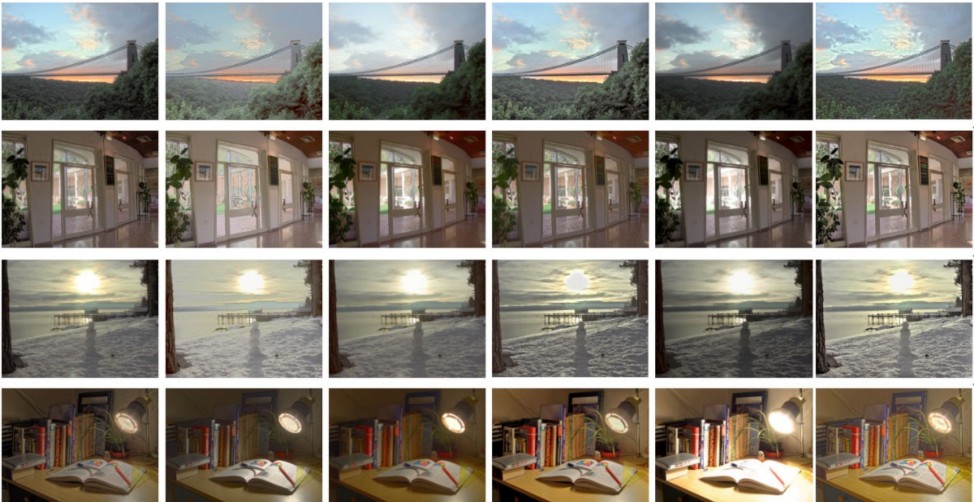

**Figure 11.** Four groups of tone-mapped images were obtained by our algorithm and the compared algorithms. The order of sub-photographs keeps in line with Figure 7.

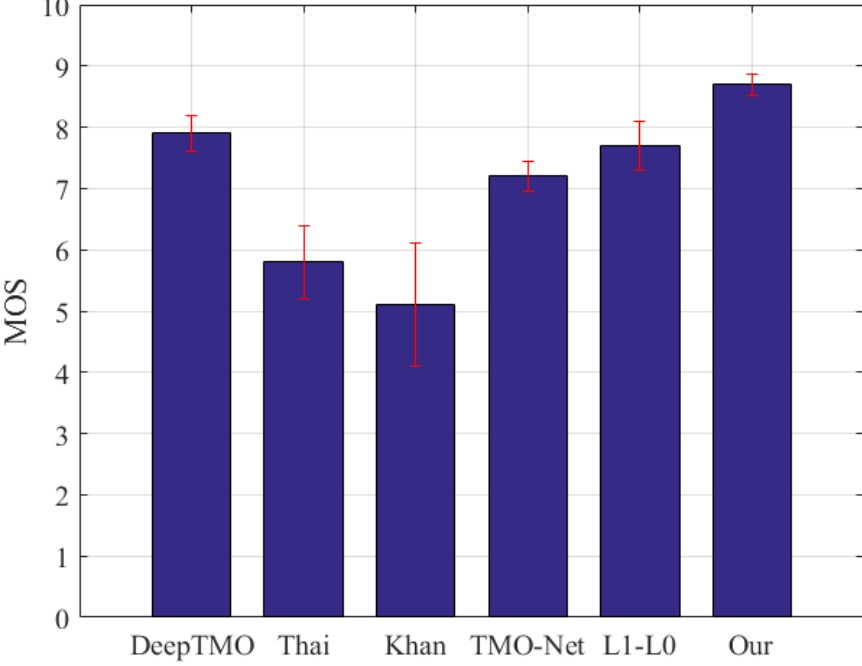

**Figure 12.** Mean and std of subjective rankings of the six tone mapping algorithms.

### 4.2. Objective Performance Evaluation

Besides subjective performance evaluation, we employed a tone-mapped image quality index (TMQI) [19] and natural image quality evaluator (NIQE) [25] as objective evaluations to further evaluate the performance of related tone mapping algorithms. The TMQI score ranges from 0 (worst quality) to 1 (best quality), and a smaller NIQE indicates a higher image quality. Tables 1 and 2 show the TMQI and NIQE scores for related tone mapping algorithms. According to Tables 1 and 2, our algorithm results in a higher score than the compared tone mapping algorithms. The mean TMQI score for our tone mapping method is higher by about 9%, 30%, 12%, 8%, and 41% as compared to DeepTMO, Thai, TMO-Net, L1-L0, and Khan tone mapping methods. The mean NIQE score for our tone mapping method is lower than the competitive tone mapping methods. The highest TMQI and lowest NIQE scores indicate the tone-mapped image obtained by our algorithm provides better structural fidelity and less statistical naturalness than the other methods.

**Table 1.** Objective evaluations using 6 tone mapping algorithms and 3 radiance maps.

| Algorithm | Memorial Church | | Moto | | Apartment | |
|---|---|---|---|---|---|---|
| | TMQI | NIQE | TMQI | NIQE | TMQI | NIQE |
| L1-L0 | 0.893 | 3.62 | 0.876 | 3.73 | 0.834 | 3.82 |
| DeepTMO | 0.905 | 3.35 | 0.904 | 3.59 | 0.798 | 3.31 |
| Khan | 0.652 | 5.29 | 0.622 | 6.21 | 0.649 | 5.45 |
| TMO-Net | 0.852 | 4.28 | 0.814 | 5.07 | 0.824 | 4.34 |
| Thai | 0.711 | 6.67 | 0.685 | 6.75 | 0.659 | 5.75 |
| Our | 0.929 | 3.21 | 0.919 | 3.42 | 0.854 | 3.03 |

**Table 2.** Objective evaluations using 6 tone mapping algorithms and 15 radiance maps from the TMQI database.

| Image Index | DeepTMO | | Thai | | TMO-Net | | L1-L0 | | Khan | | Our | |
|---|---|---|---|---|---|---|---|---|---|---|---|---|
| | TMQI | NIQE | TMQI | NIQE | TMQI | NIQE | TMQI | NIQE | TMQI | NIQE | TMQI | NIQE |
| 1 | 0.801 | 3.21 | 0.733 | 6.66 | 0.810 | 3.60 | 0.850 | 3.15 | 0.788 | 5.35 | 0.900 | 3.11 |
| 2 | 0.829 | 3.24 | 0.701 | 5.65 | 0.844 | 3.97 | 0.796 | 3.13 | 0.553 | 5.82 | 0.888 | 3.06 |
| 3 | 0.843 | 3.87 | 0.710 | 5.69 | 0.875 | 4.14 | 0.879 | 3.81 | 0.654 | 5.93 | 0.891 | 3.32 |
| 4 | 0.885 | 3.84 | 0.671 | 5.82 | 0.854 | 4.33 | 0.863 | 3.58 | 0.721 | 6.36 | 0.897 | 3.46 |
| 5 | 0.812 | 3.84 | 0.681 | 4.75 | 0.760 | 4.47 | 0.837 | 3.57 | 0.619 | 5.53 | 0.904 | 3.47 |
| 6 | 0.831 | 3.55 | 0.684 | 6.94 | 0.722 | 3.99 | 0.850 | 4.13 | 0.632 | 4.68 | 0.888 | 2.94 |
| 7 | 0.825 | 3.12 | 0.705 | 6.38 | 0.781 | 3.47 | 0.839 | 3.74 | 0.552 | 5.24 | 0.880 | 3.14 |
| 8 | 0.784 | 3.57 | 0.684 | 5.84 | 0.851 | 3.71 | 0.779 | 3.71 | 0.541 | 6.20 | 0.887 | 2.93 |
| 9 | 0.817 | 3.92 | 0.699 | 6.91 | 0.753 | 3.61 | 0.838 | 4.19 | 0.586 | 5.14 | 0.901 | 2.86 |
| 10 | 0.787 | 3.48 | 0.656 | 5.36 | 0.877 | 4.74 | 0.816 | 3.36 | 0.684 | 6.03 | 0.901 | 3.11 |
| 11 | 0.846 | 3.69 | 0.682 | 6.17 | 0.799 | 3.76 | 0.810 | 3.88 | 0.664 | 5.66 | 0.901 | 3.60 |
| 12 | 0.807 | 3.57 | 0.675 | 6.10 | 0.860 | 4.21 | 0.803 | 3.93 | 0.621 | 6.24 | 0.897 | 3.17 |
| 13 | 0.829 | 3.17 | 0.663 | 6.38 | 0.744 | 3.90 | 0.828 | 3.56 | 0.705 | 4.76 | 0.899 | 3.23 |
| 14 | 0.815 | 3.40 | 0.696 | 6.38 | 0.760 | 4.25 | 0.769 | 3.71 | 0.638 | 5.52 | 0.894 | 3.05 |
| 15 | 0.834 | 3.39 | 0.692 | 5.79 | 0.736 | 3.71 | 0.888 | 3.24 | 0.584 | 5.08 | 0.905 | 3.47 |
| Mean | 0.823 | 3.52 | 0.689 | 6.05 | 0.800 | 3.96 | 0.830 | 3.65 | 0.636 | 5.57 | 0.896 | 3.19 |

*4.3. Hardware Platform Test*

Generally, our method is purely software-based, and no specific hardware is required. Hence, our method is easily ported to many hardware platforms. In this section, our method was implemented on three hardware platforms based on ARM CPU, FPGA, and DSP architectures to analyze the feasibility of our method on embedded platforms. The ARM CPU is Qualcomm's Snapdragon 865 mobile platform. The CPU and operating system in this platform are a quadcore Kryo@ 2.84 GHz and Android 10.0. Our target FPGA platform is the Xilinx Virtex-7 XC7VX690T FPGA with 5 million gates and 52-Mbit SRAM memory for data exchange. The DSP is the DaVinci digital media processor DM648 with a 1.1 GHz maximum processing clock rate. Three implementations of the proposed method were conducted for these platforms: Java for ARM CPU, VHDL for FPGA, and C language for DSP. Table 3 summarizes the average computational time cost for different hardware platforms when running our method on the TMQI database with six kinds of resolution.

**Table 3.** The average computational time cost when running implementations on different hardware platforms.

| Resolution | $357 \times 535$ | $512 \times 380$ | $401 \times 535$ | $720 \times 480$ | $713 \times 535$ | $803 \times 535$ |
|---|---|---|---|---|---|---|
| ARM CPU | 27.24 ms | 27.75 ms | 30.60 ms | 49.29 ms | 54.41 ms | 61.28 ms |
| FPGA | 11.31 ms | 11.53 ms | 12.71 ms | 20.47 ms | 22.60 ms | 25.45 ms |
| DSP | 10.48 ms | 10.69 ms | 11.78 ms | 18.97 ms | 20.95 ms | 23.59 ms |

This is an expected result and the proposed method can be implemented on all three platforms. It can be seen that both FPGA and DSP platforms achieve real-time performance compared with ARM CPU. For the largest spatial resolution ($803 \times 535$), our method runs on FPGA and DSP platforms with a speed of about 39 FPS and 42 FPS. Since our method is very dependent on the multiplier, the runtime of the ARM CPU is markedly lower than the other platforms.

## 5. Conclusions

This paper focuses on reproducing real-world luminance on a conventional display. The radiance map is obtained by global luminance mapping and local contrast preserving. The piecewise linear mapping is introduced to achieve global dynamic range compression. For preserving local contrast, this work computes each pixel by a weighted average of its neighborhood pixels, which luminance values are known by the piecewise linear mapping. We compare the performance of our algorithm and 5 existing state-of-the-art tone mapping algorithms. Each tone mapping algorithm was performed on 3 famous HDR images and the TMQI database, respectively. The objective metric TMQI and subjective evaluation MOS are employed to evaluate every tone-mapped image. The proposed algorithm obtained the best TMQI, MOS, and NIQE among all the tone mapping algorithms, which indicates that our method can keep the quality of tone-mapped images close to the real-world scene and produce high visual quality and natural-looking images under various real-world scenes.

**Author Contributions:** Conceptualization, L.Z.; methodology, L.Z. and J.W.; software, R.S.; validation, L.Z., R.S. and J.W.; formal analysis, R.S.; investigation, R.S.; resources, J.W.; data curation, L.Z.; writing—original draft preparation, R.S.; writing—review and editing, J.W.; visualization, J.W.; supervision, L.Z.; project administration, L.Z. and J.W.; funding acquisition, L.Z. All authors have read and agreed to the published version of the manuscript.

**Funding:** This research was funded by Quzhou Science and Technology Plan Project, grant number 2022K108 and Heilongjiang Provincial Natural Science Foundation of China, grant number YQ2022F014.

**Acknowledgments:** The authors acknowledge Quzhou Science and Technology Plan Project (grant number 2022K108), Heilongjiang Provincial Natural Science Foundation of China (grant number YQ2022F014), and Basic Scientific Research Foundation Project of Provincial Colleges and Universities in Heilongjiang Province (2022KYYWF-FC05).

**Conflicts of Interest:** The authors declare no conflict of interest.

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
