# Peer review of "Three-Stage Tone Mapping Algorithm"

_electronics, doi:10.3390/electronics11244072_

Round 1

Reviewer 1 Report

The references need to be improved. Such as

https://onlinelibrary.wiley.com/doi/10.1111/cgf.13148

https://www.semanticscholar.org/paper/A-Tone-Mapping-Algorithm-for-High-Contrast-Images-Ashikhmin/976a5cf275e520b5da3b05182a01da792bac383b

https://www.researchgate.net/publication/269273856_An_improved_tone_mapping_algorithm_for_High_Dynamic_Range_images

The paper needs to be proof read more thoroughly. The authors must analyze and examine the interface and synergistic operational characteristics of their algorithm software across a variety of hardware platforms and uses. 

Author Response

Dear reviewer:

Thank you for your decision and constructive comments on my manuscript. We have carefully considered the suggestion of the Reviewer and made some changes. We have tried our best to improve and made some changes to the manuscript

The yellow part has been revised according to your comments in this paper. Revision notes, point-to-point, are given as follows:

Point 1: The references need to be improved. Such as

https://onlinelibrary.wiley.com/doi/10.1111/cgf.13148

https://www.semanticscholar.org/paper/A-Tone-Mapping-Algorithm-for-High-Contrast-Images-Ashikhmin/976a5cf275e520b5da3b05182a01da792bac383b

https://www.researchgate.net/publication/269273856_An_improved_tone_mapping_algorithm_for_High_Dynamic_Range_images

Response 1:

The three references have been referred to and cited in the article. Moreover, the index of related reference cited papers is corrected.

Point 2: The paper needs to be proof read more thoroughly. The authors must analyze and examine the interface and synergistic operational characteristics of their algorithm software across a variety of hardware platforms and uses.

Response 2:

  1. Thank you very much for your valuable comments. We add section 4.3 to analyze and examine the synergistic operational characteristics of our algorithm. In this section, our method was implemented on three hardware platforms based on ARM CPU, FPGA, and DSP architectures. Generally, our method is purely software-based and no specific hardware is required. Hence, our method is easily ported to many hardware platforms. According to the result of implementations, our method can be implemented on all three platforms. Both FPGA and DSP platforms achieve real-time performance since our method is very dependent on the multiplier.
  2. We proof read the paper and correct some mistakes both of the language and symbol.

Reviewer 2 Report

Paper no. electronics-2046456

Title: Three-Stage Tone Mapping Algorithm

In the study a tone mapping algorithm was used to to map real-world luminance into displayed luminance. The idea of the paper seems interesting and worth publication in Electronics after careful major revision.

1.       In the abstract the most important experimental results should be added.

2.       what does mean that “the third stage is very similar to the second stage”? Please correct that in the abstract.

3.       In the introduction part the novelty of the paper should be more clearly explained.

4.       All symbols in the equations should be defined.

5.       The results should be more in-depth discuss with the existing literature.

6.       Figures should be more in depth discuss in the revised version of the manuscript.  

7.       Conclusions should reflect the results obtained in the study.

Author Response

Dear reviewer:

Thank you for your decision and constructive comments on my manuscript. We have carefully considered the suggestion of the Reviewer and made some changes. We have tried our best to improve and made some changes to the manuscript

The yellow part has been revised according to your comments in this paper. Revision notes, point-to-point, are given as follows:

Point 1: In the abstract the most important experimental results should be added.

Response 1:

The quantitative results with subjective evaluation and objective evaluation have been added in the abstract, compared with the closest competitive tone mapping methods. The major visual effect of our method is mentioned in the abstract.

Point 2: what does mean that “the third stage is very similar to the second stage”? Please correct that in the abstract.

Response 2:

Thank you very much for your carefully examine. We correct the sentence in the abstract.

Point 3: In the introduction part the novelty of the paper should be more clearly explained.

Response 3:

  1. In the introduction part, we add the defect of the previous tone-curve, i.e. only compressed high luminance could cause over-compression of the global contrast. The point of our global tone mapping has been added to correcting the defect.
  2. In the introduction part, we add the problem of related local contrast preservation, i.e. insufficient performance of visual detail preservation. The point of our local contrast preservation method has been added to improve visual detail.

Point 4: All symbols in the equations should be defined.

Response 4:

Thank you very much for carefully examine. We check all the symbols in our paper and find that  equation (1) and the superscripts in equation (4) are undefined. We define the two symbols after equation (1) and equation (4). Moreover, we convert the pixel location symbols of x and y in equation (2) to subscript form. We also find the symbol  is incorrect both in equation (6) and equation (8). We replace the symbol  to  both in equation (6) and equation (8).

Point 5: The results should be more in-depth discuss with the existing literature.

Response 5:

  1. The visibility of the tone-mapped images produced by existing methods is detailed and discussed after Figure 9.
  2. We analyze the visibility of directly displaying TMQI database images on LDR display.
  3. The visibility of four groups of tone-mapping image reproduced by existing methods from the TMQI database are discussed.
  4. The mean score and standard deviation for each tone mapping method are explained after Figure 12. Meanwhile, we analyze all the mean scores and point out that our method achieves about 12% higher than the closest tone mapping method (DeepTMO).
  5. We add a detailed comparison of TMQI scores with each tone mapping method and explain that the mean TMQI score for our tone mapping method is higher by about 9%, 30%, 12%, 8%, and 41% as compared to DeepTMO, Thai, TMO-Net, L1-L0, and Khan tone mapping methods. In addition, we also explain the highest TMQI scores indicate better structural fidelity and less statistical naturalness.

Point 6: Figures should be more in depth discuss in the revised version of the manuscript.

Response 6:

  1. The explanation of Figure 2 has been added in the paper. The shape of the tone mapping curve is discussed according to Figure 1.
  2. The explanation of Figure 4 has been added after Figure 4.

Point 7: Conclusions should reflect the results obtained in the study.

Response 7:

Thank you very much for your valuable comments. The brief experimental process has been added to the conclusions. We also mention the objective metric and subjective evaluation chosen in the paper and point out that the proposed algorithm obtained both the best objective metric and subjective evaluation which means our method can keep the quality of tone-mapped images close to the real-world scene and produce high visual quality and nature-looking images under various real-world scenes.

Round 2

Reviewer 1 Report

Thanks for the responses to my review

Author Response

Thank you for your decision and constructive comments on my manuscript.

Reviewer 2 Report

The paper was corrected according my suggestions.

Author Response

(The authors gave the same response as above.)
